

# The expression of small RNAs in exosomes of follicular fluid altered in human polycystic ovarian syndrome

Junhe Hu, Tao Tang, Zhi Zeng, Juan Wu, Xiansheng Tan and Jiao Yan

Agriculture and Biotechnology Department, Hunan University of Humanities, Science and Technology, Loudi, Hunan province, China

## ABSTRACT

Polycystic ovary syndrome (PCOS) can cause reproductive disorders that may affect oocyte quality from punctured follicles in human follicular fluid (HFF). The non-coding RNA family includes micro RNA (miRNA), piwi-interacting RNA (piRNA) and transfer RNA (tRNA); these non-coding RNA transcripts play diverse functions and are implicated in a variety of diseases and health conditions, including infertility. In this study, to explore the role of HFF exosomes in PCOS, we extracted and sequenced RNA from HFF exosomes of PCOS patients and compared the analysis results with those of non-PCOS control group. The HFF exosomes were successfully isolated and characterized in a variety of ways. The sequencing results of the HFF exosomal RNA showed that about 6.6% of valid reads in the PCOS group and 8.6% in the non-PCOS group were successfully mapped to the human RNA database. Using a hierarchical clustering method, we found there were ten small RNA sequences whose expression was significantly different between the PCOS and non-PCOS groups. We chose six of them to predict target genes of interest for further GO analysis, and pathway analysis showed that the target genes are mainly involved in biosynthesis of amino acids, glycine, serine and glycosaminoglycan, as well as threonine metabolism. Therefore, the small RNA sequences contained in HFF EXs may play a key role in the mechanism that drives PCOS pathogenesis, and thereby can act as molecular biomarkers for PCOS diagnosis in the future.

## INTRODUCTION

At present, studies have found that a diverse array of hormones (FSH, LH, statins, activins and estrogens), growth factors, peptides, proteins and nucleic acids are present in human follicular fluid (HFF), and these substances help establish a unique microenvironment for the growth and maturation of oocytes (*Desrochers et al., 2016*). HFF is initially derived from the blood in blood vessels in the wall of the follicle. During development, the follicle has extensive contact with various types of adjacent cells to maintain the proper growth and development of the oocyte (*Da Silveira et al., 2012*). Studies have demonstrated that follicular somatic cells can have contact with each other, mainly through the transport of extracellular vesicles (EVs), which play important regulatory roles that can affect oocyte

Corresponding author
Junhe Hu, 260477247@qq.com

maturation. Therefore, it is likely that a large number of EVs are present in the follicular fluid in order to transport these communication signals effectively. HFF is a complex mixture of proteins, metabolites, and ionic compounds and its composition dictates the general state of follicular metabolism and therefore the maturation and quality of oocytes.

EVs are cell-derived vesicles found in the majority of eukaryotic fluids, including HFF, blood, and cell culture media. EVs can consist of several different vesicle structures and are either continuously secreted by cells or produced under certain stimulatory conditions. EVs can be broadly categorized as exosomes (EXs), microvesicles (MVs), and apoptotic vesicles (AVs) (*Raposo & Stoorvogel, 2013*). Generally, EVs are classified according to their size, with EXs ranging from 30–150 nm in size, MVs ranging from 150–1,000 nm in size, and AVs ranging from 1–5 mm in size (*Van den Brande et al., 2018*). Moreover, studies have found that EVs can contain a large number of biologically active molecular substances such as proteins, mRNAs and miRNAs, which can be released from cells via the plasma membrane by budding and fission to exert their effects on neighboring cells (*Muralidharan-Chari et al., 2010*). EXs are formed by the invagination of multivesicular bodies in the cell which then fuse to the plasma membrane and are then released outside the cell (*Desrochers et al., 2016*). EXs are typically released from vesicular structures in the endoplasmic reticulum under normal physiological conditions and are released through the plasma membrane to exert regulatory functions on nearby cells (*Farooqi et al., 2017*). The trafficking of EXs and their contents plays an important role in regulation of cell physiological functions (*Da Silveira et al., 2012*; *Di Pietro, 2016*).

It has been previously reported that EXs exist in horse (*Da Silveira et al., 2012*) and human (*Yamamoto et al., 2017*) follicular fluid under normal physiological conditions and in some cases under abnormal physiological conditions, such as in patients with polycystic ovary syndrome (*Ding et al., 2015*; *Palomba, Daolio & Sala, 2017*; *Roth et al., 2014*) and milk (*Sohel et al., 2014*). Recently, studies have reported that miRNAs can regulate the expression of various genes during oocyte maturation and follicular growth (*Hossain et al., 2012*). Several studies have reported that miRNAs are not only present in cells but also in plasma, serum, urine, saliva and milk, where they play diverse and essential biological regulatory roles (*Ding et al., 2015*; *Murri et al., 2018*; *Mitchell et al., 2008*). Additional studies have found that extracellular miRNAs are abundantly and stably present in plasma and possess high ribonuclease activity (*Van den Brande et al., 2018*) and that these miRNAs are contained within EXs and therefore protected from their external environment (*Pegtel et al., 2010*). miRNAs are small regulatory RNA molecules (typically around 22 nucleotides long) that modulate posttranscriptional gene regulation by binding to specific mRNA targets (*Pegtel et al., 2010*; *Andrade et al., 2017*). These miRNAs play essential roles in a wide range of physiological processes, many of which have a critical role in female fertility and the female reproductive system (follicular development and oocyte maturation) (*Tesfaye et al., 2018*). There have been several reports that miRNAs are present in HFF (*Da Silveira et al., 2012*; *Jiang et al., 2015*; *Liu & Zhang, 2018*), but there are no existing reports regarding HFF EXs that contain non-coding small RNA transcripts.

Polycystic ovary syndrome (PCOS) is condition that is the result of elevated androgens (male hormones) in females (approximately 5–10% of all women of reproductive age are

affected). Signs and symptoms of PCOS can include irregular or no menstrual periods, heavy periods, excess body and facial hair, acne, pelvic pain, difficulty getting pregnant, and patches of thick, darker, velvety skin (*Dewailly et al., 2010*). A previous study found that the circulating level of miRNA-21 was significantly increased in PCOS patients, which resulted in decreased expression of MST1/2, LATS1/2, TAZ in PCOS patients when compared to control subjects (*Jiang et al., 2015*). This study also showed that miR-132 and miR-320 were expressed at significantly lower levels in HFF from PCOS patients when compared to healthy controls (*Sang et al., 2013*). *Roth et al., (2014)* found that 29 miRNAs were significantly differentially expressed between PCOS and healthy control patient samples. Out of the 29 miRNAs identified in this study, hsa-miR-9, hsa-miR-18b, hsa-miR-32, hsa-miR-34c and hsa-miR-135a expression were significantly increased in PCOS follicular fluid samples compared to controls. Only miR-132 and miR-320 were found to be expressed at a significantly lower level in PCOS patients when comparing the miRNAs found in the follicular fluid of PCOS patients against that of healthy controls (*Sang et al., 2013*); this contrasts the findings regarding higher expression of miR-320 in PCOS patients in another study (*Yin et al., 2014*). The expression levels of three miRNAs, miR-222, miR-146a and miR-30c have been reported to be significantly increased in PCOS patients compared to controls; this finding was validated by the finding that miR-222 expression is positively associated with serum insulin levels while miR-146a expression is negatively associated with serum testosterone levels (*Long et al., 2014*). Although some studies have reported alterations in MiRNA expression in PCOS, there are no existing reports regarding HFF EXs that contain non-coding small RNA transcripts and PCOS or reports of MiRNAs that may be promising candidates for biomarkers for PCOS diagnosis and treatment.

It is likely that EXs and their contents, such as miRNAs, play a vital regulatory role in HFF during oocyte growth and development. Therefore, it is necessary to compare the expression levels of non-coding small RNA transcripts in HFF-derived EXs from PCOS and healthy patients. The findings of this study revise and improve our understanding of the content of HFF-derived EXs, thus laying the foundation for the future investigation of the role of miRNAs in PCOS pathogenesis.

# MATERIAL AND METHODS

## Follicular fluid sample collection

The patients that donated the follicular fluid samples used in this study were undergoing conventional IVF treatment. Moreover, all experiments were approved by the Institutional Ethics Committee of Hunan University of Humanities, Science and Technology (# 20180308). Informed consent was obtained from each couple regarding the use of the follicular fluid sample that was obtained during IVF treatment oocyte retrieval for this study, which was approved by the Institutional Ethics Committee of Shaoyang HuiEn Reproductive and Health Hospital (#20180316). The form is attached in a Supplemental File. The current diagnostic criterion for PCOS is based on the revised 2003 criteria (two out of three is enough for positive diagnosis) as following: (a) oligo-ovulation and/or anovulation; (b) clinical and/or biochemical signs of hyperandrogenism; and (c) polycystic ovaries.

**Table 1  Clinical characteristics of PCOS and control subjects.**

| Sample name | P1 | P2 | C1 | C2 |
| --- | --- | --- | --- | --- |
| BMI (kg/m$^2$) | 24.9 | 26.8 | 23.9 | 26.3 |
| Age (years) | 29 | 40 | 45 | 43 |
| Anti-Müllerian hormone (ng/mL) | 15.88 | 8.42 | 1.59 | 2.69 |
| Antral Follicle count (number) | 20 | 16 | 17 | 14 |
| Luteinizing hormone (mIU/mL) | 11.6 | 4.25 | 2.57 | 5.1 |
| Follicle stimulating hormone (mIU/mL) | 5.14 | 5.59 | 4.86 | 5.58 |
| Estradiol (pg/mL) | 51.3 | 42.65 | 31.81 | 33.31 |
| Prolactin (ng/mL) | 11.87 | 6.14 | 14.66 | 8.14 |
| Free testosterone (ng/mL) | 62.15 | 42.37 | 41.68 | 33.36 |

Controls and PCOS patients (patient basic information in Table 1) were injected with recombinant FSH following treatment with GnRH agonists according to the standard protocol for IVF treatment. FSH stimulation was initiated once downregulation was confirmed by ultrasound and measurements of serum estradiol, luteinizing hormone, and progesterone. Real-time ultrasound scans were used to assess follicular growth at two day intervals from day 5 of FSH treatment until the day of oocyte retrieval. When at least one ovarian follicle had grown to 18–20 mm in diameter, the ovums were punctured under the guidance of vaginal B-ultrasound after 34–38 h following hCG trigger.

HFF was collected via transvaginal ultrasound-guided puncture and aspiration of follicles that were 18–20 mm in diameter. HFF was collected from each patient donor. HFF samples were centrifuged at 1,300 g for 15 min to remove the cells; blood and other material obtained was then stored at −80 °C for further experiments.

## Exosomes purification and characterization

HFF EXs were purified and characterized according to previously published protocols with some modifications (*Cecilia, Maria & Jan, 2012*). The EX isolation method was performed as follows: a 15 mL pooled sample generated from combined HFF from patients was centrifuged at 3,500 rpm for 15 min at 4 °C to pellet debris. The supernatant was then transferred to a 15 mL ultracentrifuge tube and ultracentrifuged at 16,500 g for 30 min at 4 °C, and then filtered through a 0.2 mm syringe filter to obtain the medium contained exosomes. Finally, exosomes were pelleted by ultracentrifugation at 120,000 g for 70 min at 4 °C and stored at −80 °C for further analysis.

For nanoparticle tracking analysis (NTA), obtained exosomes were diluted in phosphate-buffered saline (PBS). Specifically, samples were diluted with sterile PBS according to the manufacturer's instructions. Capture and analysis settings were manually set according to the above clinical protocol. Using a Flow Nano-Analyzer instrument (NanoFCM Inc., Xiamen, China), EXs were visualized with laser light scattering, and Brownian motion of the EXs was captured on video. Recorded videos were further analyzed with Image J computer software, which provided high-resolution particle size distribution profiles and concentration measurements of the EXs in solution.

Flow cytometry analysis was carried out using a previously published protocol (*Wahlgren et al., 2012*). Briefly, stored EX pellets are resuspended with 100 ul of filtered PBS and

kept on ice. A negative control sample was prepared without EX staining and labeled NC, and the positive sample was stained with antibodies for CD63 and CD81 and labeled accordingly. Samples and controls were suspended in PBS and analyzed at a FACS workstation (NanoFCM Inc., Xiamen, China) with the guidance of the instrument operating procedures.

EX pellets were suspended in PBS for electron microscopy analysis. First, 5–10 μL of a resuspended EXs pellet is added to a copper mesh and precipitated for 3 min as filter paper absorbs the volatile liquid from the edge. Second, negative staining with phosphotungstic acid was carried out following a PBS wash. Finally, the mesh was dried at room temperature for 2 min before imaging on the electron microscope operating voltage 80–120 kv (JEM-1200EX, Japan Electronics Co., Ltd).

### EX small RNA library construction

EX pellets were resuspended in Trizol (Invitrogen) for RNA isolation. The small RNA library from EXs was constructed using the total RNA from 15 mL of pooled HFF. Small RNA cloning, sequencing, and analysis were carried out as described previously using the QIAseq® miRNA Library Kit (*Sang et al., 2013*). RNA detection and quality control was carried out using an Agilent 2100 Bio-analyzer (Agilent Technologies Sweden AB).

### RNA-seq data analysis of EXs from human follicular fluids

Initially, the sequencing adaptors and low-complexity reads were removed in an initial data filtering step, and the quality of reads was estimated with the FASTQC program. Reads were aligned against the human reference genome (hg19 February 2009, GRCh37) downloaded from the GENCODE project (http://www.gencodegenes.org/) and used for all subsequent bioinformatic analyses. Finally, we applied a DEBseq-counts algorithm to filter the differentially expressed genes following FDR analysis using the criteria Log2FC>1 and FDR<0.05 to choose the significantly expressed genes for further research. Functional annotation was performed using the Database for Annotation, Visualization and Integrated Discovery (DAVID) v6.8 (https://david.ncifcrf.gov). Pathway analysis was carried out using annotated data downloaded from KEGG (https://www.genome.jp/kegg/).

### Statistical analysis

Both the back-spliced junction reads and linear mapped reads were combined and scaled to reads per kilobase per million mapped reads (RPKM) to quantify the expression level of miRNAs, piRNAs, and tRNAs. Differences in expression profiles between the PCOS group and the non-PCOS group were analyzed using a Student's $t$-test. A $P < 0.05$ was considered to indicate statistically significant differences.

## RESULTS

### Isolation and characterization of HFF-derived EXs

HFF EXs were successfully separated and characterized as described in the methods section. EXs isolated from HFF had an average diameter of 75 nm, which is consistent with the reported range of EX sizes, as shown in Fig. 1A. Moreover, CD63 and CD81 are enriched

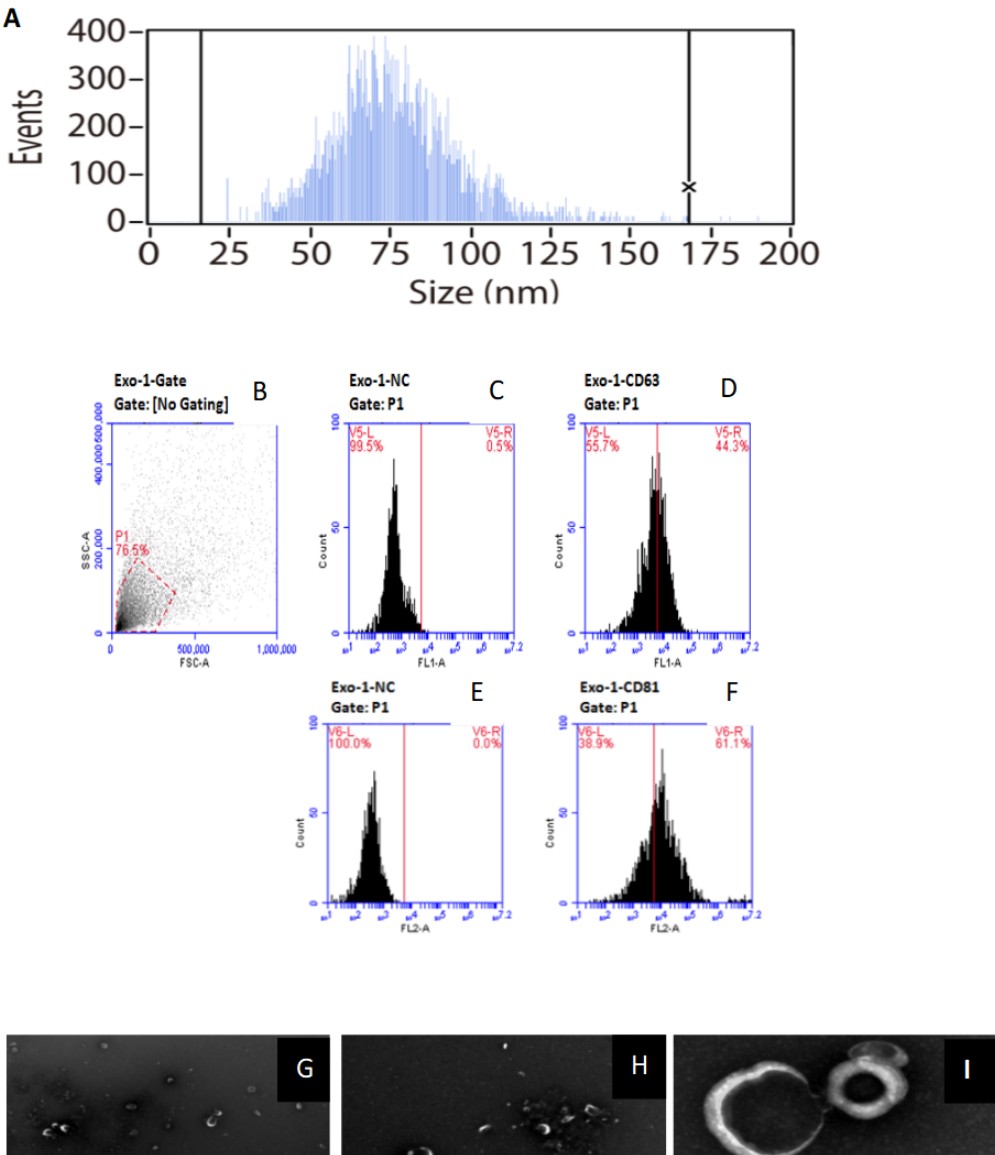

**Figure 1 Characterization of of HFF exosomes.** (A) Nanoparticle tracking measurement of exosomes particle size and concentration was shown. Pelleted fractions from FF are vesicles whose diameter size ranged from 15 to 165 nm, with a peak size between 50 and 100 nm. Average vesicle size was 75 nm. Particle size is consistent with exosome size range. (B) Flow cytometry without checking samples. (C) Flow cytometry picture for exosomes sample without using monoclonal antibodies against the tetras panins CD63. (D) Flow cytometry picture for exosomes sample using monoclonal antibodies against the tetras panins CD63. (E) Flow cytometry picture for exosomes sample without using monoclonal antibodies against the tetras panins CD81. (F) Flow cytometry picture for exosomes sample using monoclonal antibodies against the tetras panins CD81. (G) Transmission electron microscopy of exosomes under the view of 2 um size microscope rulers. (H) Transmission electron microscopy of exosomes under the view of 500 nm size microscope rulers. (I) Transmission electron microscopy of exosomes under the view of 200 nm size microscope rulers.

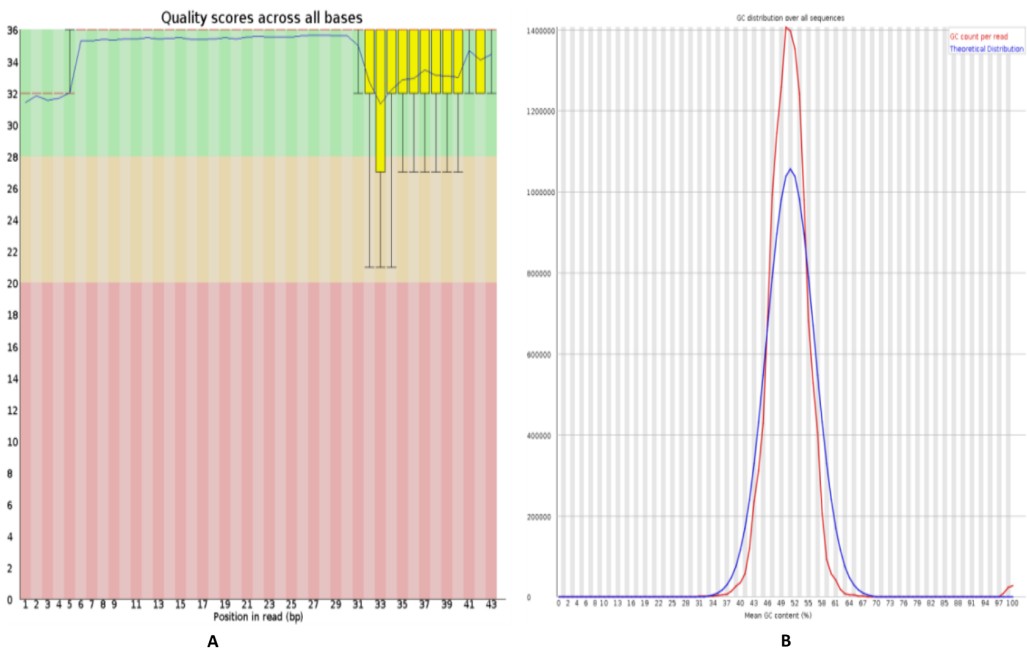

**Figure 2 Quality control of RNA-Seq data of HFF exosomes.** (A) The horizontal axis represents the number of bases or the range of the number of bases, and the vertical axis represents the value of the quality score, where Quality scores for 20 indicates that the mapping accuracy is greater than 99% and can be used for subsequent analysis. (B) The horizontal axis represents average Quality scores and the vertical axis represents reads. The blue line indicates the distribution of theoretical GC content, and the red line indicates the distribution of actual GC content.

in EXs, and our samples were positive for both, as shown in Figs. 1B–1F. EX pellets were resuspended in PBS for electron microscopy analysis, as shown in Figs. 1G–1I, to further confirm the presence of intact EXs.

## Quality control analysis of RNA-seq data

FastQC software (https://www.bioinformatics.babraham.ac.uk/projects/fastqc/) was used to perform evaluate the quality of sequenced data, including mass value distribution, position value distribution, and the GC content, as shown in Fig. 2. The quality control analysis of the raw sequencing data gives a quick overview of the data before use in further analysis.

## Analysis of small RNAs in HFF EXs

RNA sequencing and analysis were carried out using four EX samples (two prepared from PCOS patients compared against two prepared from non-PCOS patients). On average, the total number of valid reads obtained from the EXs of patients with PCOS (7.50 million) was similar that of the control EXs (6.86 million), as shown in Table 2. Overall, about 6.6% and 8.6% of all valid reads from this set of two PCOS and two non-PCOS EX samples, respectively, could be successfully mapped to the human non-coding RNA database and studied further (Table 2).

**Table 2** The results of showing non-coding RNAs sequencing data mapping to the human genomes from HFF exosomes.

| Sample name | P1 | P2 | C1 | C2 |
|---|---|---|---|---|
| All Reads | 9,163,276 | 5,827,574 | 8,657,970 | 5,070,271 |
| UnMapped | 8,835,532 | 5,269,328 | 7,946,685 | 4,611,534 |
| Mapped | 327,744 | 558,246 | 711,285 | 458,737 |
| Mapped Rate | 0.036 | 0.096 | 0.082 | 0.09 |
| Unique Mapped | 250,217 | 398,550 | 562,175 | 344,625 |
| Unique Mapped Rate | 0.027 | 0.068 | 0.065 | 0.068 |
| Repeat Mapped | 77,527 | 159,696 | 149,110 | 114,112 |

MiRNAs differentially expressed between the PCOS and non-PCOS samples were further analyzed using hierarchical clustering methods to generate a heatmap comparing the results for the PCOS group (sample P1 and P2) and control group (sample C1 and C2), as shown in Fig. 3A. From this analysis comparing the PCOS and control groups, we chose ten significantly upregulated (miR-6087, miR-4745-3p, miR-193b-3p, miR-199a-5p, miR-4532, miR-199a-3p, miR-199b-3p, miR-629-5p, miR-143-3p, and miR-25-3p) and 10 significantly downregulated (miR-98-5p, miR-483-5p, miR-382-5p, miR-23b-3p, miR-10a-5p, miR-200a-3p, miR-141-3p, miR-3911, miR-200c-3p, and miR-483-3p) miRNAs for further study, as shown in Fig. 3B. Analysis of differently expressed piRNAs was also performed using hierarchical clustering methods to generate a heatmap comparing the PCOS group (sample P1 and P2) and the control group (sample C1 and C2), as shown in Fig. 4A. From this analysis comparing the PCOS and control groups, we chose ten piRNAs with significantly upregulated expression (pir-36441, pir-57942, pir-54998, pir-34896, pir-33221, pir-51671, pir-33226, pir-43997, pir-33405, and pir-36040) and ten piRNAs with significantly downregulated expression (pir-43772, pir-35414, pir-43771, pir-35413, pir-35469, pir-33065, pir-35463, pir-35468, pir-35467, and pir-33387) for further study, as shown in Fig. 4B. Analysis of differentially expressed tRNAs was also performed using hierarchical clustering methods to generate a heatmap comparing the PCOS group (sample P1 and P2) and the control group (sample C1 and C2), as shown in Fig. 5A. We chose ten tRNAs with significantly upregulated expression (tsrna-12365, tsrna-12363, tsrna-12362, tsrna-12364, tsrna-12361, tsrna-12360, tsrna-12359, tsrna-17099, tsrna-17100, and tsrna-12395) and ten tRNAs with significantly downregulated expression (tsrna-06177, tsrna-06176, tsrna-14935,tsrna-14937, tsrna-15209, tsrna-14934, tsrna-15199, tsrna-15198, tsrna-03939, and tsrna-03940) for further study, as shown in Fig. 5B.

## GO and pathway analysis of miRNA target genes

Mature miRNAs are produced by a series of nueclease-mediated cleavages of longer primary transcripts, which are then assembled into RNA-induced silencing complexes in order to identify target genes via complementary base pairing. This guides the silencing complexes to degrade target genes or block the translation of target genes according to the degree of complementarity with the miRNA sequence.

Based on the above research results, six miRNAs (miR-6087, miR-199a-5p, miR-143-3p, miR-483-5p, miR-200a-3p, and hsa-miR-23b-3p) were predicted to be relevant to

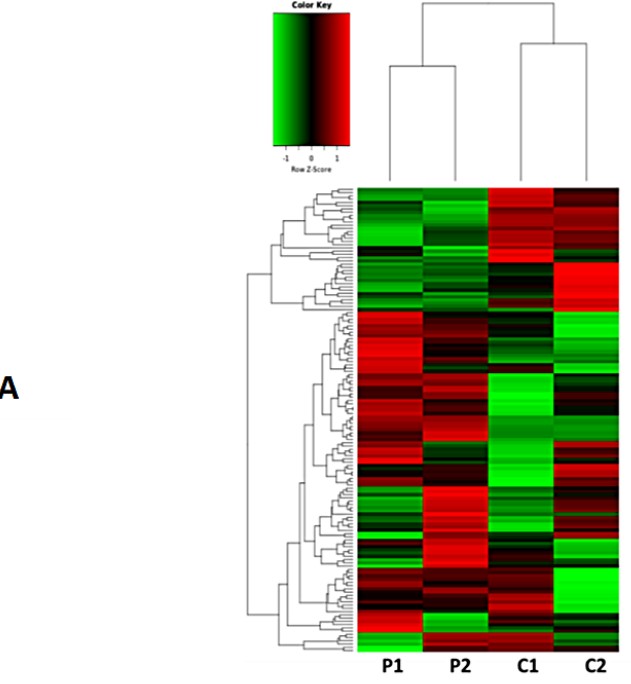

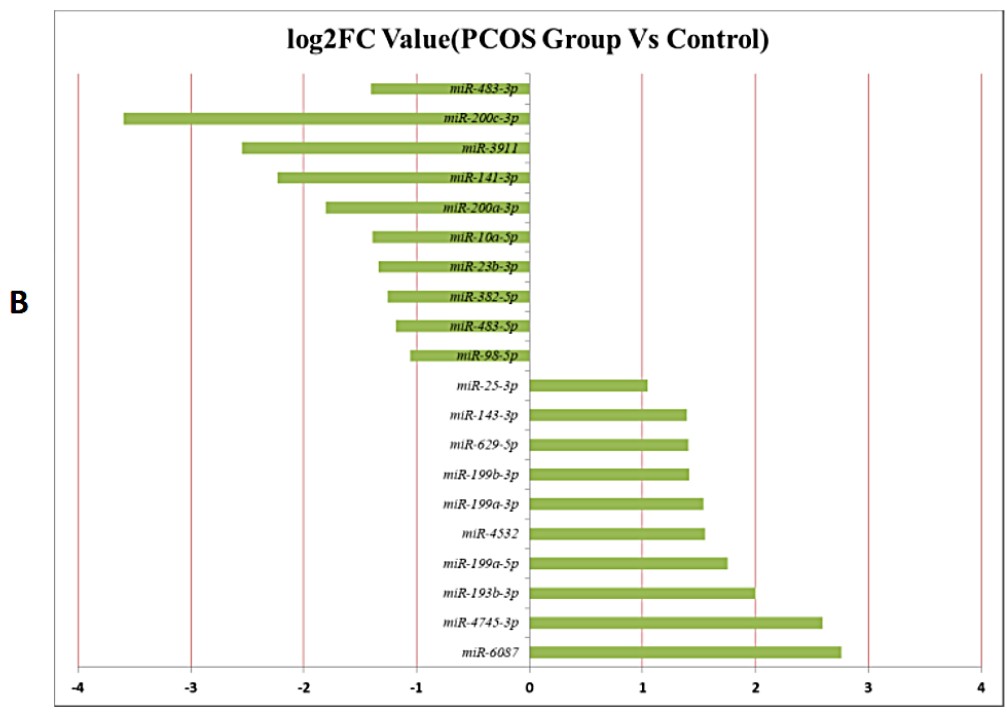

**Figure 3** **The miRNA expression analysis of HFF exosome.** (A) Heat map of hierarchical clustering analysis of miRNA shown in HFF of PCOS group (P1 and P2) and the control group (C1 and C2). (B) These are 10 significantly up and down expression miRNA separately according to their $P$ value and fold changes ($P < 0.05$ and log 2—(fold-changes)—>1).

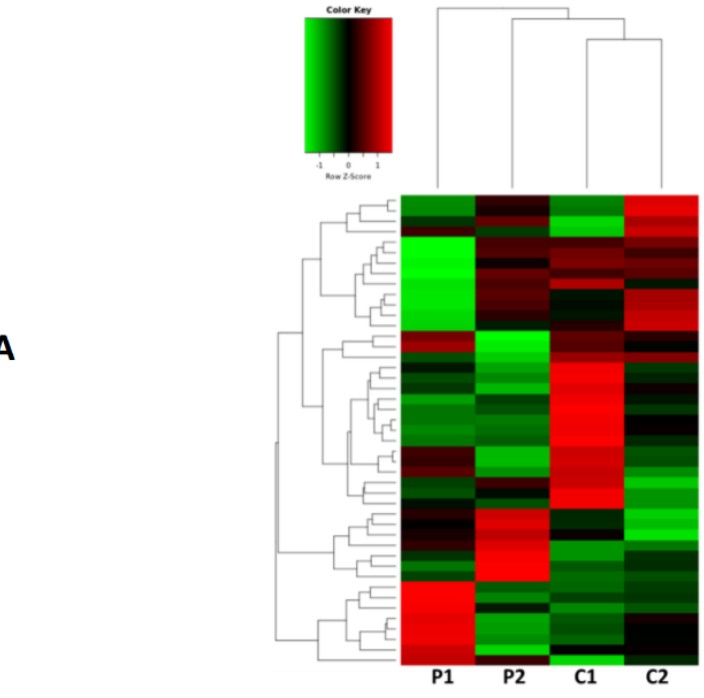

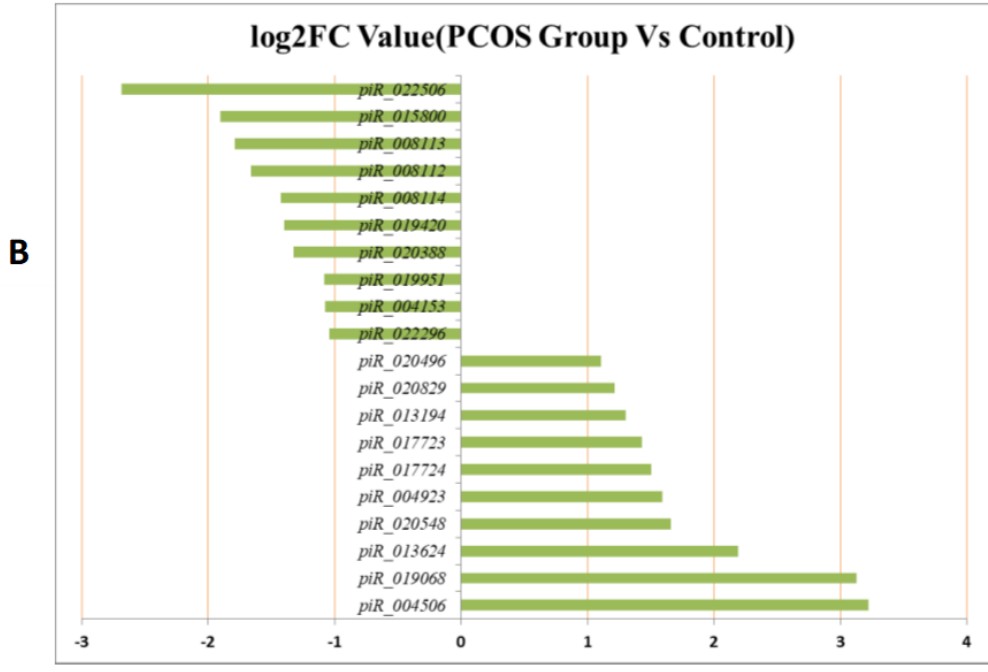

**Figure 4** **The PiRNA expression analysis of HFF exosomes.** (A) Heat map of hierarchical clustering analysis of PiRNA shown in HFF of PCOS group (P1 and P2) and the control group (C1 and C2). (B) These are 10 significantly up and down expression PiRNA separately according to their $P$ Value and fold changes ($P < 0.05$ and log 2—(fold- changes)—>1).

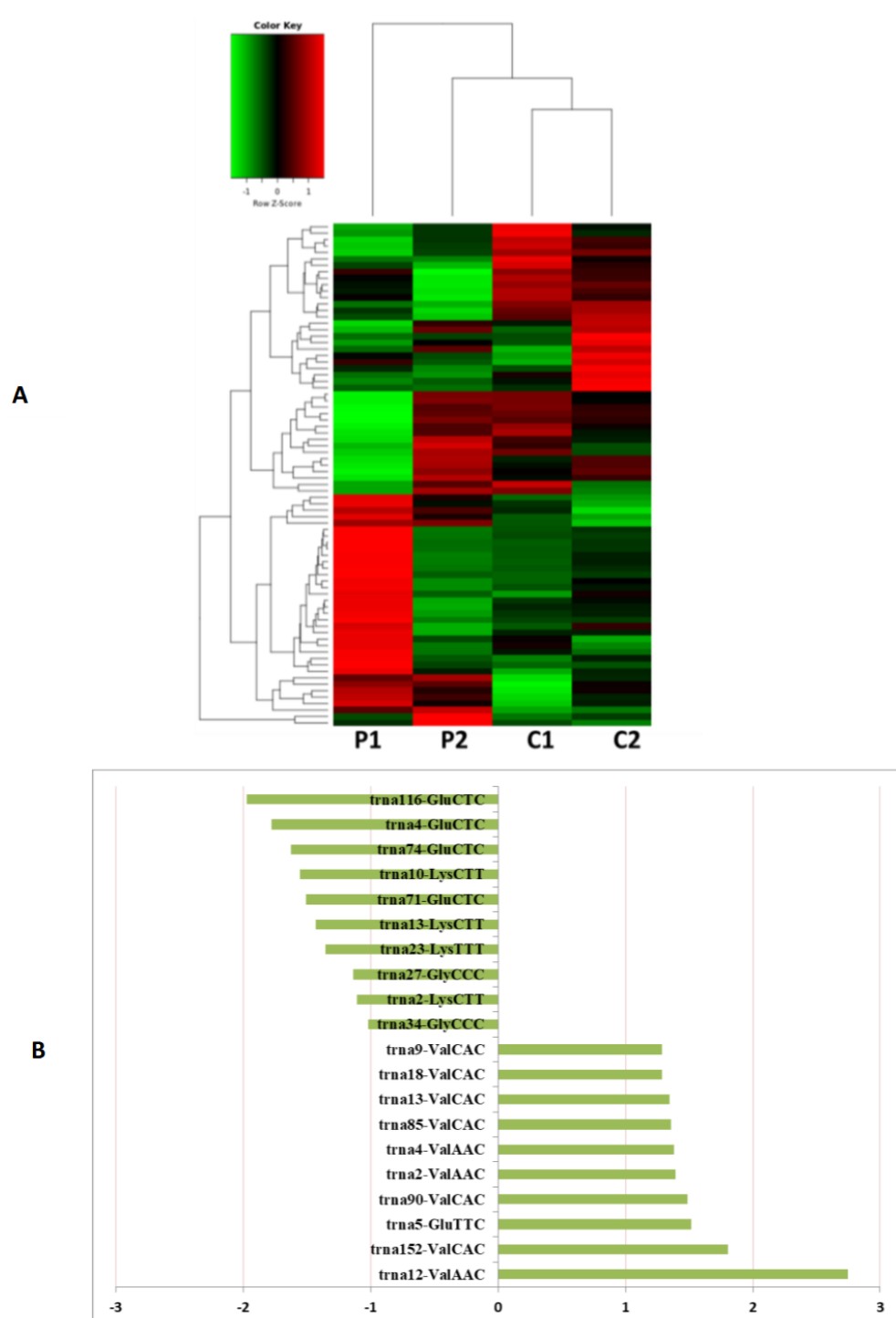

**Figure 5 The tRNA expression analysis of HFF exosomes.** (A) Heat map of hierarchical clustering analysis of PiRNA shown in HFF of PCOS group (P1 and P2) and the control group (C1 and C2). (B) These are 10 significantly up and down expression tRNA separately according to their $P$ Value and fold changes ($P < 0.05$ and log 2—(fold-changes)—>1).

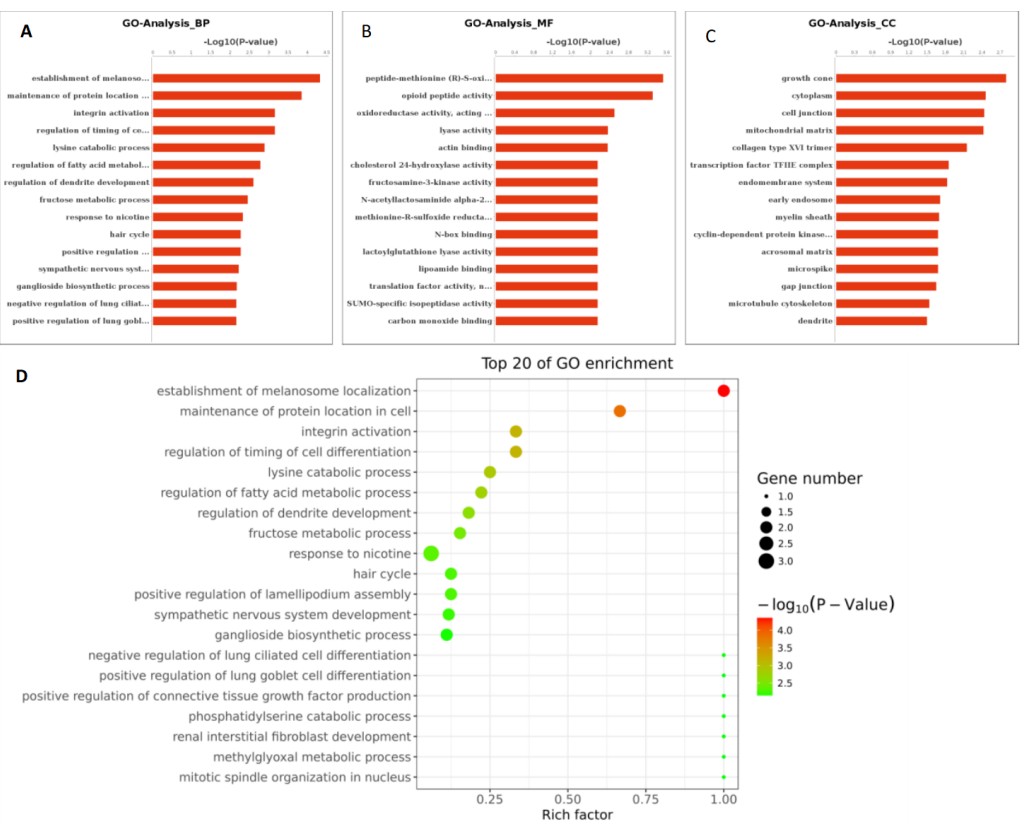

**Figure 6 GO function analysis of different expression genes.** (A) GO analysis of all the differentially expressed genes included the leftmost Biological Process (BP), the intermediate Molecular Function (MF), and the rightmost Cellular Component (CC). (B) The scatter plot is a graphical display of GO enrichment analysis results. The degree of GO enrichment is measured by Rich factor, *P* value and the number of genes enriched in this pathway. Rich factor refers to the ratio of the number of differentially enriched genes to the number of annotated genes in GO. The larger the Rich factor, the greater the degree of enrichment.

human PCOS when Miranda and RNA software were used to predict the targeting genes regulation relationship (the intersecting results of the two prediction softwares were used as the final list of target gene predictions). The parameters of this analysis were set as follows: energy_miranda <−20, score_miranda >150, and energy_RNAhybird <−25. The results of this analysis reveal 146 significantly different target genes, which are used to take GO further as shown in Fig. 6 and pathway analysis as shown in Fig. 7.

## DISCUSSION

The follicular follicle in the ovary provides a unique microenvironment for the maturation of oocytes and interactions between follicular somatic cells and oocytes. It is therefore crucial to dissect the molecular components of HFF in order to elucidate the mechanisms and regulatory processes involved in oocyte maturation. EXs are known to be essential carriers for signal transduction-mediated interaction between follicular somatic cell sand oocytes in order to promote oocyte maturation in HFF (*Cakmak et al., 2016*; *Van Blerkom,*

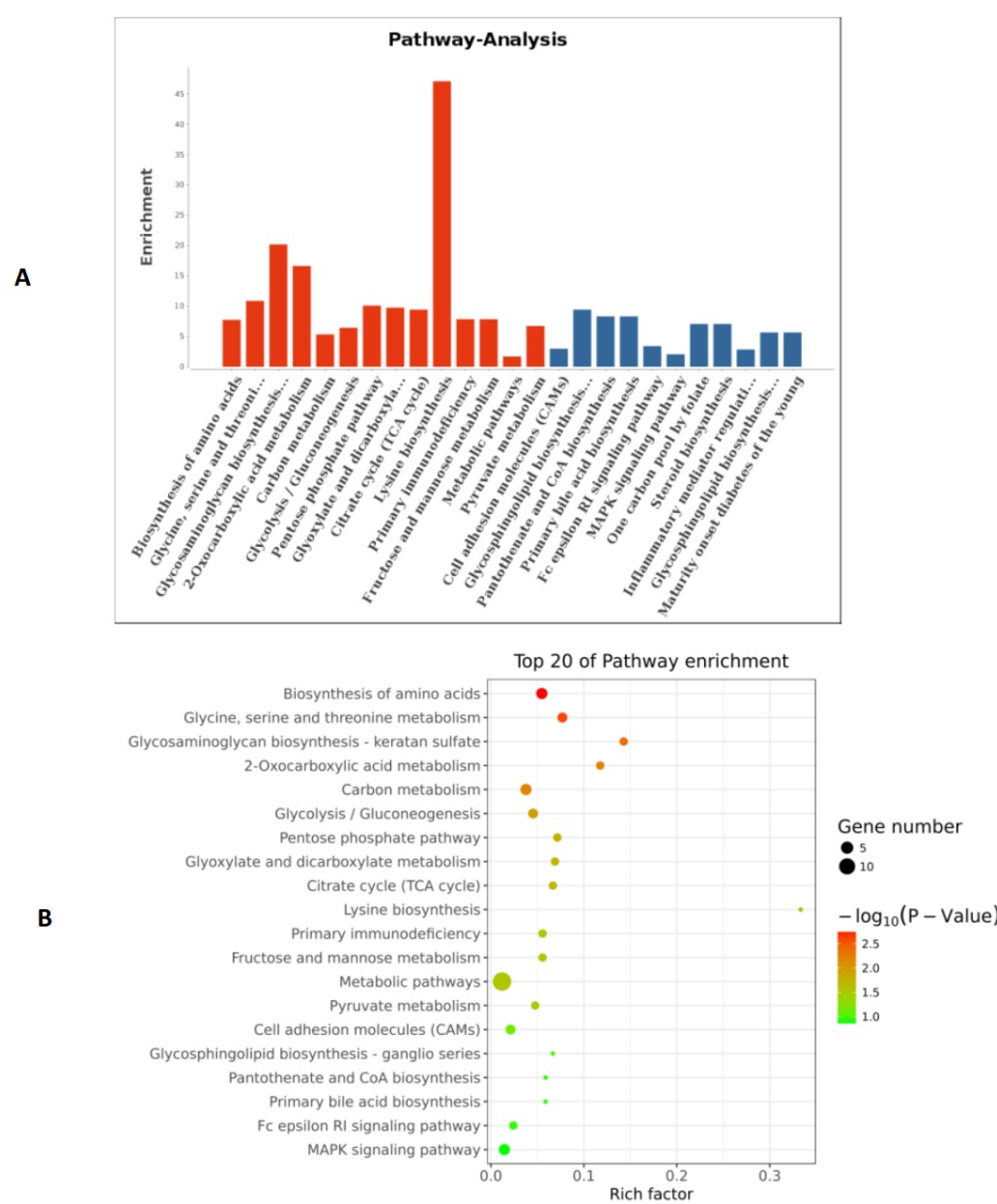

**Figure 7 The pathways of the top 20 entries.** (A) Coordinate axis $Y$: Pathway-Term entry name; coordinate axis X: log10 ($P$-Value). Red represents salient items and blue represents non-salient items. (B) The scatter plot is a graphical display of the results of pathway enrichment analysis. Pathway enrichment is measured by Rich factor, $P$ value and the number of genes enriched in this pathway. Rich factor refers to the ratio of the number of differentially enriched genes to the number of annotated genes in the Pathway. The larger the Rich factor, the greater the degree of enrichment.

*2012*). In this study, we found that HFF EXs can be successfully isolated and have an average diameter of 75 nm. The presence of intact isolated EXs was confirmed using transmission electron microscopy and FACS (EXs are positive for CD63 and CD81) (*Chen et al., 2016*; *Lo Sicco et al., 2018*; *Yang et al., 2017*).

Additional studies of the miRNAs contained and expressed in HFF EXs are necessary to reveal the mechanisms that promote oocyte maturation and how this process is regulated by the axis of follicular somatic cells-EXs-oocytes. Similarly to Sang et al. (*Brännström et al., 2010*), we identified potential EX-contained miRNAs and their gene targets using bioinformatics approaches. We found that about 6.6% of PCOS and 8.6% of non-PCOS patient EX-contained miRNAs could be successfully mapped to the human miRNA database (Table 2), revealing ten miRNAs with significantly upregulated expression (miR-6087, miR-4745-3p, miR-193b-3p, miR-199a-5p, miR-4532, miR-199a-3p, miR-199b-3p, miR-629-5p, miR-143-3p, and miR-25-3p) and ten miRNAs with significantly downregulated expression (miR-98-5p, miR-483-5p, miR-382-5p, miR-23b-3p, miR-10a-5p, miR-200a-3p, miR-141-3p, miR-3911, miR-200c-3p, and miR-483-3p) when comparing results from PCOS and healthy patients. Several of these miRNAs are potentially novel biomarkers for PCOS that have not been mentioned in previous reports (*Murri et al., 2013*; *Sathyapalan et al., 2015*). Furthermore, we found a variety of piRNAs and tRNAs are significantly differentially expressed when comparing results from PCOS and healthy patients (Figs. 4 and 5). PiRNAs are small RNA molecules that are 24–32 nucleotides long and are abundant in the germline across animal species  (*Thomson & Lin, 2009*). A previous study identified twenty-six piRNAs that were differentially expressed in the cumulus cells of diminished ovarian reserve patients compared to controls (*Chen et al., 2017*). We identified ten piRNAs whose expression is either significantly up- or downregulated in PCOS patients compared to controls (Fig. 4). Some of these piRNAs could be potential biomarkers for molecular diagnosis of PCOS in the future. Our results also show that ten tRNAs whose expression is either significantly different up- or downregulated in PCOS patients compared to controls (Fig. 5). It is interesting to note that mitochondrial tRNA mutations have been previously associated with PCOS (*Ding et al., 2018*). Some of these significantly differentially expressed small RNAs could play mechanistic roles PCOS pathogenesis and should be studied farther.

Through our analysis, we identified six miRNAs (miR-6087, miR-199a-5p, miR-143-3p, miR-483-5p, miR-23b-3p and miR-200a-3p) that could be relevant to PCOS pathogenesis. These six MiRNAs underwent target genes analysis, which revealed that miR-483-5p is reported as PCOS biomarker (*Shi et al., 2015*), as well as miR-200a-3p (*Dhaded & Dabshetty, 2018*). Currently, there is no gold standard for miRNA gene target analysis (*Zhang, Zhang & Su, 2009*), so we elected to use Miranda and RNA software to take GO analysis and pathways according to reported papers (*Riffo-Campos, Riquelme & Brebi-Mieville, 2016*). The results of our analysis show that the relevant miRNA-regulated pathways are mainly related to amino acid biosynthesis, glycine, serine and threonine metabolism, glycosaminoglycan biosynthesis, monocarboxylic acid metabolism, and carbon metabolism. However, these potential miRNA targeted genes and pathways are predicted, not experimentally validated, and therefore the associations observed in this study are limited. Some studies have shown that each miRNA can have hundreds of gene targets and the gene targets may be different based on cell type and the presence or absence of other miRNAs (*Zhang, Zhang & Su, 2009*). The impact of a given miRNA on gene expression may be physiologically essential but difficult to identify with bioinformatics alone (*Ricardo et al., 2009*). To date, there is no diagnostic test for PCOS, only several classes

of diagnostic criteria from 3 different entities: the Rotterdam criteria (*Lauritsen et al., 2014*), NIH criteria (*Ricardo et al., 2009*), and Androgen Excess and PCOS Society (*Abbott & Fida, 2013*). Consequently, diagnosing PCOS is highly dependent on the symptomatic criteria used (*Yildiz et al., 2012*). Although PCOS is the most common endocrine disorder in women, few susceptibility genes have been consistently linked with the syndrome (*Anvesha et al., 2013*).

Therefore, our study revealed that various miRNAs, piRNAs and tRNAs are differentially expressed in EXs in the HFF of women with PCOS when compared to controls, and these small RNAs target gene clusters and pathways that correspond to a variety cell functions. Some of these target genes may be potential biomarkers for PCOS diagnosis and treatment. Moreover, many questions remain unanswered regarding the relationship between MiRNAs in EXs and the mechanisms that drive PCOS, including the following: what is the relationship between the small RNAs (miRNA, piRNA and tRNA) contained in HFF EXs and PCOS pathogenesis? Which miRNAs might promote the development of PCOS? Do small RNAs contained in HFF EXs play a mechanistic role in driving PCOS?

## CONCLUSIONS

We found that a variety of non-coding small RNAs contained in HFF-derived EXs from PCOS patients are significantly differentially expressed when compared to control patient samples. Some of these differentially expressed small RNAs or their target genes may be potential biomarkers for diagnosis and treatment of PCOS. However, the mechanism that relates HFF EX-contained miRNAs to PCOS pathogenesis remains to be elucidated in future studies.

### Funding

This work is fully funded by the Hunan Natural Science Foundation (2019JJ40139). The funders had no role in study design, data collection and analysis, decision to publish, or preparation of the manuscript.

### Grant Disclosures

The following grant information was disclosed by the authors:
Hunan Natural Science Foundation: 2019JJ40139.

### Competing Interests

The authors declare there are no competing interests.

### Author Contributions

- Junhe Hu conceived and designed the experiments, performed the experiments, analyzed the data, prepared figures and/or tables, authored or reviewed drafts of the paper, and approved the final draft.
- Tao Tang and Jiao Yan performed the experiments, authored or reviewed drafts of the paper, and approved the final draft.

- Zhi Zeng analyzed the data, prepared figures and/or tables, and approved the final draft.
- Juan Wu performed the experiments, analyzed the data, prepared figures and/or tables, and approved the final draft.
- Xiansheng Tan conceived and designed the experiments, analyzed the data, prepared figures and/or tables, authored or reviewed drafts of the paper, and approved the final draft.

## Human Ethics

The following information was supplied relating to ethical approvals (i.e., approving body and any reference numbers):

The Institutional Ethics Committee of Hunan University of Humanities, Science and Technology approved this research (#20180308).

## Field Study Permissions

The following information was supplied relating to field study approvals (i.e., approving body and any reference numbers):

The Institutional Ethics Committee of Shaoyang HuiEn reproductive and health Hospital approved this research (#20180316).

## Data Availability

Data is available at the National Genomics Data Center: CRA001999.

## Supplemental Information

Supplemental information for this article can be found online at http://dx.doi.org/10.7717/peerj.8640#supplemental-information.

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
