# Peer review of "The expression of small RNAs in exosomes of follicular fluid altered in human polycystic ovarian syndrome"

_PeerJ, doi:10.7717/peerj.8640_

## Round 0.1 · original submission · Major Revisions

Three specialists in the field evaluated the present manuscript, and they have concerns related to this paper. The reviewers have described all points that should be answered by the authors. Please ensure that the English language in this submission meets our standards: uses clear and unambiguous text, is grammatically correct, and conforms to professional standards of courtesy and expression. Considering the evaluation carried out by all reviewers, I recommend major revision in this submission.

Reviewer 1 ·

Basic reporting

see last page.

Experimental design

nothing to add

Validity of the findings

see last page.

Additional comments

First of all , your abstracts needs more detail. I suggest that you improve the description at line 11 ”functionally diverse,including infertility” change to “functionally diverse in lots of diseases, including infertility”. And line 17 ”6.6% and 8.6%” change to “6.6% in PCOS and 8.6% in non-PCOS group ”.

The next most important item is your introduction. I suggest that you improve the description at lines 48”secreted by cells” change to “secreted out of the cells”. Besides,at line 50, the “Necrotic debris (NDs)”, please explain it, I couldn't find it in reference 3. If it's irrelevant, you should delete it. And it'll correspond with the next sentence. What’s more, at line 71, the “high ribonuclease activity”, please point it out, as I couldn't find it in reference 15,16, and I am curious about it.

Most importantly , the English language should be improved to ensure that an international audience can clearly understand your text. Some examples where the language could be improved include lines 59, 61,62, the current phrasing makes comprehension difficult.Especially at line 62,”which have target cell signaling functions by containing various RNAs” is better. Moreover, at line 84, the ”enhanced” word, it should change to “increased”. At first, “enhanced” usually describe qualilty instead of quantity, secend, “increased” correspond with “decreased”.

Last, I assume that “34-8” which at line 126 is just a slip of the pen.

I commend the authors for their extensive data set, compiled over many years of detailed fieldwork. In addition, the manuscript is clearly written in professional, unambiguous language.

Reviewer 2 ·

Basic reporting

no comment

Experimental design

1- data for samples is lost. please data table of a sample characteristic

Validity of the findings

please compare in a biomarker for this research with older literature.
Are there common ncRNAs between the different study groups?

Additional comments

1- data for samples is lost. please data table of a sample characteristic
2- please compare in a biomarker for this research with older literature. Are there common ncRNAs between the different study groups?

Reviewer 3 ·

Basic reporting

See general comments section

Experimental design

See general comments section

Validity of the findings

See general comments section

Additional comments

Polycystic ovarian syndrome (PCOS) is one of the most common causes of infertility in women, which is due to failure of ovulation and excessive ovarian androgen production. The follicular fluid (FF) in the ovary play an important role in the maturation of oocytes. The follicular somatic cell and oocyte interact with each other in FF through exosomes (EXs) during oocyte maturation. In this paper, Hu et al. isolated exosomes from FF of PCOS patients and analysed their small RNAs content to find a novel biomarker for PCOS diagnosis. Although this paper contains valuable scientific data, the presentation of the paper needs to be improved.
1. The title of the paper is unclear, which should be changed so that it can reflect the outcome of the study. For example, the title could be “The expression of small RNAs in exosomes of follicular fluid altered in human polycystic ovarian syndrome”.
2. Although it is informative, the abstract is difficult to read. So, the abstract should be re-written in such a way that it is easy to understand
3. The introduction is relevant to the study carried out in the paper. However, it is not clear and therefore it should be re-written so that the reader can understand what is known and what is unknown in the field.
4. Methods are described with sufficient detail and the results section is clear.
5. The discussion section can be cut down by removing the repetition of results (for example Fig. 3 results in the 2nd paragraph).
6. There are several typos and grammatical mistakes in the manuscript that require attention.

---

## Round 0.2 · Minor Revisions

One of the reviewers asked for additional corrections in the abstract. Please read the reviewer's comments.

Reviewer 3 ·

Basic reporting

See general comments section

Experimental design

See general comments section

Validity of the findings

See general comments section

Additional comments

Many of my comments are addressed in the revised manuscript. However, the abstract is still unclear and there are several typos and grammatical mistakes that require attention.

---

## Round 0.3 · accepted · Accept

The authors carried out all modifications requested by the reviewers in each phase of the reviewing process of this manuscript. In my view, the paper improved and can be accepted as it is.